# Risk Factors for Postsurgical Gout Flares after Thoracolumbar Spine Surgeries

**DOI:** 10.3390/jcm11133749

**Published:** 2022-06-28

**Authors:** Kuan-Jung Chen, Yen-Chun Huang, Yu-Cheng Yao, Wei Hsiung, Po-Hsin Chou, Shih-Tien Wang, Ming-Chau Chang, Hsi-Hsien Lin

**Affiliations:** 1Department of Orthopedics, China Medical University Hsinchu Hospital, Hsinchu 302, Taiwan; ronald96016@gmail.com; 2Department of Orthopedics and Traumatology, Taipei Veterans General Hospital, Taipei 112, Taiwan; fu6294613@gmail.com (Y.-C.H.); orthycyao@gmail.com (Y.-C.Y.); edward.kuma@gmail.com (W.H.); choupohsin@gmail.com (P.-H.C.); stwang@vghtpe.gov.tw (S.-T.W.); mcchang@vghtpe.gov.tw (M.-C.C.); 3School of Medicine, National Yang Ming Chiao Tung University, Taipei 112, Taiwan

**Keywords:** gout, flare, postsurgical gout flare, spine surgery, risk factor

## Abstract

Gouty arthritis is the most common form of inflammatory arthritis and flares frequently after surgeries. Such flares impede early patient mobilization and lengthen hospital stays; however, little has been reported on gout flares after spinal procedures. This study reviewed a database of 6439 adult patients who underwent thoracolumbar spine surgery between January 2009 and June 2021, and 128 patients who had a history of gouty arthritis were included. Baseline characteristics and operative details were compared between the flare-up and no-flare groups. Multivariate logistic regression was used to analyze predictors and construct a predictive model of postoperative flares. This model was validated using a receiver operating characteristic (ROC) curve analysis. Fifty-six patients (43.8%) had postsurgical gout flares. Multivariate analysis identified gout medication use (odds ratio [OR], 0.32; 95% confidence interval [CI], 0.14–0.75; *p* = 0.009), smoking (OR, 3.23; 95% CI, 1.34–7.80; *p* = 0.009), preoperative hemoglobin level (OR, 0.68; 95% CI, 0.53–0.87; *p* = 0.002), and hemoglobin drop (OR, 1.93; 95% CI, 1.25–2.96; *p* = 0.003) as predictors for postsurgical flare. The area under the ROC curve was 0.801 (95% CI, 0.717–0.877; *p* < 0.001). The optimal cut-off point of probability greater than 0.453 predicted gout flare with a sensitivity of 76.8% and specificity of 73.2%. The prediction model may help identify patients at an increased risk of gout flare.

## 1. Introduction

Gouty arthritis is the most common form of inflammatory arthritis, with a worldwide prevalence of 2.49–6.24% [1,2,3]. Hospital admission and surgical treatment are both established risk factors for acute flare-ups of gouty arthritis [4]. Such flares lead to longer hospital stays and increased healthcare expenses [5,6]. In the postoperative period, these flares impede early patient mobilization. Therefore, patients with known risk factors should be identified and given optimal medical treatment preoperatively.

The rate of acute postoperative flare-ups in patients with a history of gouty arthritis ranges from 17.2% to 44.3% [7,8,9,10,11]. The possible precipitating factors for gout flare associated with surgery include starvation, volume depletion, and tissue hypoxia [12]. Previous studies have identified presurgical uric acid levels ≥9 mg/dL and the extent of change in uric acid levels postoperatively as risk factors for postsurgical flares [9,11].

In orthopedics surgeries, the spine procedures were among the most complex categories, with relatively more blood loss and longer surgical duration. In our clinical experience, postsurgical gout flares frequently occur after thoracolumbar spine procedures in patients with a medical history of gouty arthritis. However, the exact flare rate has rarely been reported. Whether spinal surgeries share the same risk factors as general surgeries is unclear. Moreover, in spine surgeries, decompression, instrumentation, and fusion each involve different degrees of bone resection, burring, curettage, and drilling. The association between the extent of surgery and triggering of a gout flare is also unknown. Moreover, as minimally invasive spine (MIS) surgeries have become popular in recent years, it remains unclear whether patients with a gout history who underwent MIS procedures have a decreased rate of gout flare.

In this study, we evaluated the risk factors and clinical characteristics of gout flares after thoracolumbar spine surgeries and aimed to construct a predictive model of postoperative gout flares.

## 2. Materials and Methods

This was a retrospective study of patients prospectively collected in a database of a single medical center from 2009 to June 2021. The database inclusion criteria were consecutive patients aged ≥18 years who underwent thoracolumbar spine surgery for degenerative etiologies under general anesthesia; the database comprised 6439 patients. Additional criteria specific to the current investigation included a history of gouty arthritis identified by the registry of the International Classification of Disease, Tenth Revision, Clinical Modification (ICD-10-CM, codes M10.0-M10.9) and a minimum of 30 days’ follow-up. Patients who underwent surgery for malignant metastasis, infectious spondylodiscitis, and acute high-energy trauma were excluded. Patients who experienced gout flares preoperatively were also excluded. For patients who underwent more than one operation within the study period, only data from the first admission were analyzed. The study was conducted according to the guidelines of the Declaration of Helsinki, and approved by the relevant institutional review board, which waived the requirement of obtaining informed consent for this study due to its retrospective nature.

Baseline demographics included age, sex, body mass index (BMI), comorbidities, medications for gout control (including colchicine or urate-lowering therapy), diuretic use, smoking habits, and alcohol use. An alcohol intake habit was defined as more than two drinks per day for men and one drink per day for women [13]. Operative details, including operative time, type of surgery, number of operated levels, use of instrumentation, use of fusion, MIS procedure, and revision surgery, were collected from medical charts. By definition, procedures performed with a paraspinal approach and less muscle stripping (minimally invasive transforaminal lumbar interbody fusion [MI TLIF], microdiscectomy, etc.) were considered MIS procedures. Procedures performed with a conventional posterior approach (open TLIF, posterior lumbar fusion, instrumentation of dynamic stabilization devices, etc.) were not considered to be MIS procedures. Perioperative details, including intraoperative blood loss and fluid intake, intraoperative and postoperative transfusion, preoperative hemoglobin (Hb) level, and Hb level on the first postoperative day were also recorded. The decrease in Hb level on the first postoperative day compared with the Hb level preoperatively was calculated.

Medical records were searched for gout flares within 30 days postoperatively. Gout flares were confirmed by consultation with rheumatologists within 24 h of an attack. Laboratory tests, imaging assays, or arthrocentesis were performed at the discretion of rheumatologists according to the guidelines for diagnosis of the American College of Rheumatology [14,15]. In patients with postoperative gout flare, the postoperative day of onset, number and site of the involved joint, and uric acid level at the onset of gout flare were collected from medical charts.

### Statistical Analyses

Patients were categorized into two groups based on whether they experienced a gout flare within 30 days postoperatively: the flare-up and no-flare groups. Statistical analyses were performed using MedCalc version 20 (MedCalc Software bv, Ostend, Belgium). Categorical data were compared using the chi-square or Fisher’s exact test. Continuous data were accessed for normality with the Shapiro–Wilk test and compared using an independent t-test or Wilcoxon rank-sum test, as appropriate. A *p*-value < 0.05 was considered statistically significant. For factors with significant differences between groups, odds ratios (ORs) and their respective 95% confidence intervals (CIs) were calculated using multivariate logistic regression analysis. A prediction model was constructed using significant factors determined by multivariate logistic regression analysis. Probability were calculated for every patient in the dataset. A receiver operating characteristic (ROC) curve was plotted, and the area under the curve (AUC) was evaluated. The optimal cut-off point for predicting gout flare postoperatively was determined using the Youden index.

## 3. Results

A total of 132 patients had a history of gouty arthritis within the database of 6439 patients (prevalence, 2.1%). After excluding patients who underwent fixation for high-energy trauma (1 patient) or malignant metastasis (1 patient), or who had flare-up preoperatively (2 patients), 128 patients were finally enrolled in the study.

In total, 102 (79.7%) men and 26 (20.3%) women (average age, 64.8 years) were included. The flare-up group included 56 patients (43.8%), while the no-flare group included the remaining 72 patients. No inter-group difference was found in terms of age, sex, or BMI.

The flare-up group had a significantly lower rate of medical control of gouty arthritis (20/56, 35.7% vs. 46/72, 63.9%, *p* = 0.002), higher rate of smoking (30/56, 53.6% vs. 22/72, 30.6%, *p* = 0.01), and lower rate of hyperlipidemia (5/56, 8.9% vs. 17/72, 23.6%, *p* = 0.03) than the no-flare-up group. Moreover, the length of hospital stay was longer in the flare-up group than in the no-flare group (12.9 vs. 8.6 days, *p* = 0.01; Table 1).

Operative factors specific to spine surgeries are listed in Table 2 and did not differ between groups. Intraoperative blood loss, fluid intake, and intraoperative and postoperative transfusion rates did not differ between groups. However, pre- and postoperative Hb levels were significantly lower in the flare-up group than in the no-flare-up group (12.8 vs. 13.6 g/dL, *p* = 0.02 and 10.5 vs. 11.8 g/dL, *p* = 0.001, respectively). Additionally, decreases in Hb level on the first postoperative day were also higher in the flare-up group than in the no-flare-up group (2.2 vs. 1.8 g/dL, *p* = 0.03).

A multivariate logistic regression analysis revealed that the predictors for postsurgical gout flare were smoking, use of gout medications, preoperative Hb level, and Hb drop. Since pre- and postoperative Hb levels were highly related, the latter was not introduced in the multivariate analysis. After accounting for confounding variables, hyperlipidemia was not found to be a factor of postsurgical flares (Table 3).

A prediction formula was constructed using multivariate logistic regression based on significant factors, including smoking, gout medication use, preoperative Hb level, and Hb drop, as follows: $$P\left(Y=1\right)=\frac{e^{3.8\;-\;1.2\mathrm{Med}\;+\;1.3\mathrm{smoking}\;-\;0.4\mathrm{preOP}\;\mathrm{Hb}\;+\;0.7\mathrm{Hb}\;\mathrm{drop}}}{1+e^{3.8\;-\;1.2\mathrm{Med}\;+\;1.3\mathrm{smoking}\;-\;0.4\mathrm{preOP}\;\mathrm{Hb}\;+\;0.7\mathrm{Hb}\;\mathrm{drop}}}\;$$

Refer to the Appendix A for the calculator of this prediction formula.

An ROC curve was created by calculating the probability predicted with this formula for every patient in the dataset. The AUC was found to be 0.801 (95% CI, 0.717–0.877, *p* < 0.001). The optimal cut-off point of probability (0.453) had a sensitivity of 76.8% and a specificity of 73.2% (Figure 1).

Gout flares occurred within a mean of 3.5 (range, 1–22) days, postoperatively, with the highest number of gout flares recorded on the second postoperative day (Figure 2). The mean uric acid level upon the onset of gout flare was 6.2 ± 2.0 mg/dL. More cases with monoarticular involvement (26/47, 55.3%) than cases with involvement of two or more joints were recorded. Most flare patterns involved only the joints in the lower extremities (40/47, 85.1%). Only three patients presented with flare-ups limited to the upper extremities (6.4%), while four patients had flare-ups in both the upper and lower extremities (8.5%). The most frequently involved joints were the knee joints (31/47, 65.9%), ankle joints (24/47, 51.1%), and first metatarsophalangeal joints (7/47, 14.9%) (Table 4).

## 4. Discussion

The prevalence of gouty arthritis in our patient database was 2.1% in adult patients admitted for spine surgery. We found that 43.8% of patients with gout history experienced postsurgical gout flares. A history of smoking, lack of medical gout control, lower preoperative Hb level, and greater decrease in Hb level on the first postoperative day were risk factors for postsurgical gout flare.

The overall prevalence of gouty arthritis is 3.9% in the US [2] and 2.49% in the UK [1]. The 2.1% prevalence found in the current study was lower than that in the previous study for several reasons. First, our institute is located in an urban area, with a lower prevalence rate of gouty arthritis reported [3]. Second, the specific population who underwent spine surgery may differ from the general population in terms of the prevalence rate of gouty arthritis. Third, we utilized the ICD-10-CM codes to identify patients with a diagnosis of gouty arthritis; therefore, we may not have detected those with a missed registry.

The flare-up rate of gouty arthritis, postoperatively, was reported to be 17.2–44.3% [7,8,9,10,11], which was comparable with the 43.8% flare-up rate observed in this study. In previous studies, cancer and abdominal surgeries were found to be risk factors for gout flare-up [8,11]. To the best of our knowledge, the flare-up rate after spine surgery has not been reported in the literature. Similar to previous findings in general surgery, the current study found that age, sex, BMI, diuretic use, comorbidity, and operative duration were not associated with postsurgical gout flares [8,9,11]. We further investigated surgical factors specific to spine surgery, including the type of spine surgery, number of operated levels, instrumentation procedures, fusion procedures, and revision surgeries. However, none of these factors were associated with postsurgical gout flares. Although studies have demonstrated that MIS techniques lead to lower infection rates, shorter hospital stays, and better patient-reported outcomes than conventional open techniques [16,17], we did not find a difference in the rate of MIS techniques between the flare-up and no-flare groups. A similar finding was described by Zhou et al., who reported that the rates of open and endoscopic surgery were comparable between the flare-up and no-flare groups in a cross-sectional study of 518 patients [8]. This suggests that factors other than the extent of surgery are involved.

In this study, we found that a lower preoperative Hb level and a greater Hb level decrease on the first postoperative day were risk factors for postoperative gout flare. Anemia has been found to cause impaired oxidative metabolism, which may lead to hyperuricemia and gout onset [18,19,20]. Moreover, excessive bleeding leads to volume depletion and metabolic acidosis, which promote urate crystallization [21]. Blood transfusion and IV fluid were also found to be associated with in-hospital gout flare in general hospitalized patients [22]. Although intraoperative blood loss, fluid intake, and blood transfusions were also analyzed in this study and in several previous studies, none of these factors were found to be associated with gout flares [8,9,11]. This reflects the fact that a postoperative decreased level of Hb is the net result of an intricate interaction between the estimated blood loss, fluid administration, transfusions, and hemostatic status of each patient.

In our study, smoking was a risk factor for postsurgical gout flares. Zhou et al. also found a higher rate of smoking in the flare-up group than in the no-flare group. However, the difference did not reach statistical significance [8]. A study demonstrated that smoking temporarily decreased the serum uric acid levels, which returned to the initial level within hours after smoking [23]. This may explain our findings, as the hospitalized patients were forced to abstain from smoking, which resulted in fluctuations in serum uric acid levels and provoked gout flares. In addition to smoking, alcohol has long been recognized as a risk factor for gout attacks [24,25]. However, no significant association was found between alcohol and gout flares in our study. This may be due to the small number of cases that met the defined criteria for habitual alcohol use in both groups.

The effect of medical control of gouty arthritis in postsurgical flares was noted in the current study, as well as in several previous studies. In this series, we found an OR of 0.324 for gout flares in patients receiving regular medications for gouty arthritis. Moreover, Jeong et al. reported an OR of 0.11 for gout flares in patients receiving regular medications for gouty arthritis in their multivariate analysis [9]. Two studies investigated the effect of colchicine prophylaxis, reporting ORs of 0.071 and 0.157 [8,11]. However, the dosage of the medication was not specified in these studies. We may refer to a multidisciplinary consensus that suggested prophylactic administration of colchicine according to creatinine clearance [26]. Based on the findings of this study, the probability of gout flare in each patient can be calculated with the prediction formula. For probability above the cut-off level of 0.453, the model predicted a gout flare with a sensitivity of 76.8% and a specificity of 73.2%. Thus, in such individuals, prophylactic colchicine may be appropriate.

The most common pattern of joint involvement in post-spinal surgery gout flares was monoarticular (55.3%) and lower extremities (85.1%). The results of this study were comparable to postsurgical flare after general surgeries, with 46–71% monoarticular involvement and 84–97% lower extremity involvement [7,8,9,11]. The mean onset of gout flare in this study was at 3.5 days postoperatively, with a mean uric acid level of 6.2 mg/dL during the flare, similar to that previously reported (3.7–4.2 days postoperatively and 6.0–6.4 mg/dL) [8,9,11]. Studies have demonstrated that patients with a greater postoperative decrease in uric acid levels were associated with gout flare [8,9,11]. However, since preoperative uric acid levels are not routinely tested in patients, we were unable to analyze this factor.

The limitations of our study included its retrospective design and the relatively small number of patients. Second, we used the ICD-10-CM codes to identify patients with gouty arthritis. There is a possibility that some patients with a history of gout may have not been identified. Third, the preoperative uric acid level, which is a known risk factor for postsurgical gout flare, was not obtained. Finally, since this is the first study to report risk factors of gout flares after spine surgeries, the prediction formula was validated internally. We require future prospective studies to further validate or modify the prediction formula.

## 5. Conclusions

Gout flares frequently occur after thoracolumbar spine surgery in patients with a medical history of gouty arthritis. Smoking, use of gout medications, preoperative Hb level, and Hb level decrease were factors predicting postoperative flare. The prediction formula allows us to calculate the probability of gout flare in individuals, and those with calculated probability higher than 0.453 may be considered appropriate candidates for prophylactic colchicine administration.

## Figures and Tables

**Figure 1 jcm-11-03749-f001:**
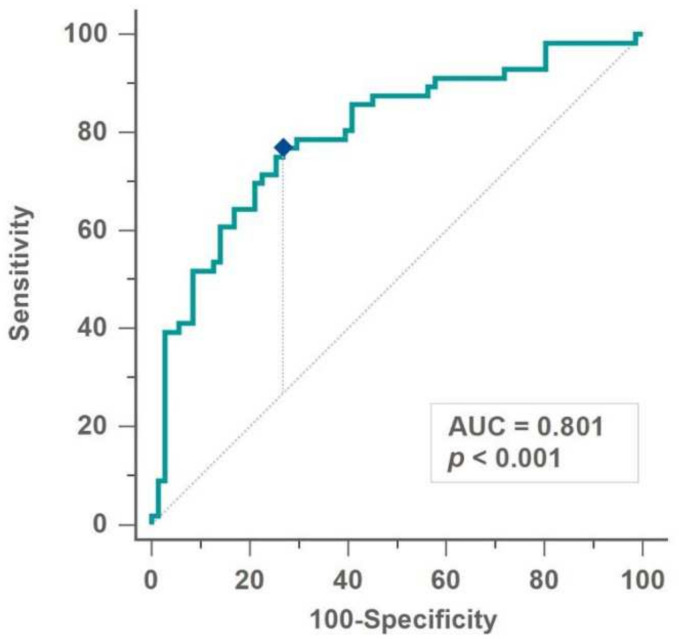
ROC curve of the probability calculated by the prediction formula. The optimal cut-off point (diamond mark) yielded a sensitivity of 76.8% and a specificity of 73.2%.

**Figure 2 jcm-11-03749-f002:**
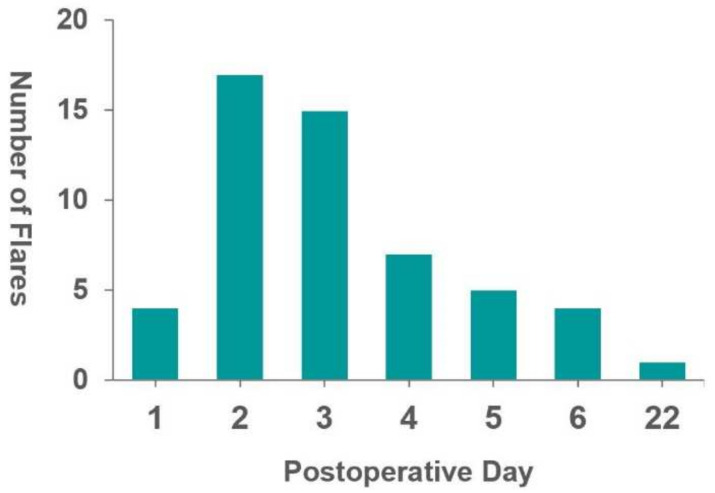
Number of gout flares by postoperative day.

**Table 1 jcm-11-03749-t001:** Demographic characteristics of patients.

	All (*n* = 128)	Flare-Up (*n* = 56)	No Flare (*n* = 72)	*p* Value
Age (years)	64.8 ± 11.7	66.2 ± 9.8	63.9 ± 13.0	0.27
Sex (men:women)	102:26	48:8	54:18	0.10
BMI (kg/m^2^)	27.8 ± 4.3	28.4 ± 4.4	27.1 ± 4.0	0.11
Comorbidities (number of patients)				
Hypertension	86 (67%)	37 (66%)	49 (68%)	0.81
Diabetes mellitus	37 (29%)	16 (29%)	21 (29%)	0.94
Hyperlipidemia	22 (17%)	5 (9%)	17 (24%)	0.03 *
Cardiovascular disease	22 (17%)	9 (16%)	13 (18%)	0.77
Chronic kidney disease	7 (5%)	2 (4%)	5 (7%)	0.41
COPD	5 (4%)	3 (5%)	2 (3%)	0.45
Malignancy	6 (5%)	2 (4%)	4 (6%)	0.70
Gout medication use (number of patients)	66 (52%)	20 (36%)	46 (64%)	0.002 *
Benzbromarone	28 (22%)	9 (16%)	19 (26%)	
Allopurinol	15 (12%)	4 (7%)	11 (15%)	
Febuxostat	13 (10%)	3 (5%)	10 (14%)	
Colchicine	12 (9%)	4 (7%)	8 (11%)	
Sulfinpyrazone	1 (1%)	1 (2%)	-	
Probenecid	1 (1%)	-	1 (1%)	
Diuretic use (number of patients)	23 (18%)	8 (14%)	15 (21%)	0.34
Smoking (number of patients)	52 (41%)	30 (54%)	22 (31%)	0.01 *
Alcohol use (number of patients)	4 (3%)	1 (2%)	3 (4%)	0.63

BMI = body mass index, COPD = chronic obstructive pulmonary disease, * Statistically significant.

**Table 2 jcm-11-03749-t002:** Operative and Perioperative factors of patients.

	All (*n* = 128)	Flare-Up (*n* = 56)	No Flare (*n* = 72)	*p* Value
Operative time (min)	235 ± 98	237 ± 94	235 ± 106	0.90
Type of surgery				0.80
Decompression or discectomy	13 (10%)	6 (11%)	7 (10%)	
Instrumentation without fusion	20 (16%)	8 (14%)	12 (17%)	
Instrumentation & MISS fusion	10 (8%)	3 (5%)	7 (10%)	
Instrumentation & open fusion	85 (66%)	39 (70%)	46 (64%)	
Number of operated levels	3.3 ± 1.1	3.3 ± 1.0	3.3 ± 1.2	0.90
Number of operations with instrumentation	115 (90%)	50 (89%)	65 (90%)	0.83
Number of operations with fusion	92 (72%)	40 (71%)	52 (72%)	0.98
MISS procedure	15 (12%)	4 (7%)	11 (15%)	0.25
Revision surgery	20 (16%)	9 (16%)	12 (17%)	0.93
Intraoperative blood loss (mL)	620 ± 430	626 ± 389	616 ± 461	0.90
Intraoperative fluid intake (mL)	2316 ± 1235	2369 ± 1232	2279 ± 1258	0.73
Intraoperative transfusion (number of patients)	55 (43%)	25 (45%)	31 (43%)	0.86
Postoperative transfusion (number of patients)	16 (13%)	10 (18%)	6 (8%)	0.12
Preoperative Hb (g/dL)	13.2 ± 1.9	12.8 ± 2.1	13.6 ± 1.8	0.02 *
Postoperative Hb (g/dL)	11.2 ± 1.8	10.5 ± 1.8	11.8 ± 1.7	0.001 *
Hb level decrease on the first postoperative day (g/dL)	2.1 ± 1.2	2.2 ± 1.2	1.8 ± 1.0	0.03 *

MISS = minimally invasive spine surgery, Hb = hemoglobin, * Statistically significant.

**Table 3 jcm-11-03749-t003:** Results of multivariate logistic regression analyses.

Variables	Regression Coefficients	Odds Ratio	95% Confidence Interval	*p* Value
Gout medication use	−1.23	0.32	(0.14, 0.75)	0.009 *
Smoking	1.31	3.23	(1.34, 7.80)	0.009 *
Hyperlipidemia	−0.09	0.53	(0.19, 1.44)	0.213
Hb, preoperative	0.66	0.68	(0.53, 0.87)	0.002 *
Hb level decrease	−0.41	1.93	(1.25, 2.96)	0.003 *

* Statistically significant.

**Table 4 jcm-11-03749-t004:** Clinical characteristics of patients with postsurgical gout flare.

Variables	Values
Postoperative day of onset (days) (mean and range)	3.5 (1–22)
Number of involved joint (number) (%)	
Monoarticular	26 (46)
Oligo or polyarticular	21 (38)
Unspecified	9 (16)
Involved joints at flare (number) (%)	
Lower extremity	40 (71)
Upper extremity	3 (5)
Both upper & lower extremities	4 (7)
Unspecified	9 (16)
Flare site (number) (%)	
Knee	31 (55)
Ankle	24 (43)
1st MTP joint	7 (13)
Foot except 1st MTP joint	5 (9)
Wrist	5 (9)
Elbow	3 (5)
Hand	1 (2)
Uric acid level on gout flare (mg/dL)	6.2 ± 2.0

MTP, metatarsophalangeal.

## Data Availability

The data presented in this study are available in this article. Further datasets of this study are available from the corresponding author on reasonable request.

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
