# Peer review of "Risk Factors for Postsurgical Gout Flares after Thoracolumbar Spine Surgeries"

_jcm, 2022, doi:10.3390/jcm11133749_

Round 1

Reviewer 1 Report

The study examined factors associated with postsurgical gout flare.

Overall impression: This is a concise and well-structured manuscript. The content and findings are not novel but are still considered valuable for clinicians. There are however some issues with methods and discussion that should be addressed before the manuscript could be considered ready for publication.

Specific comments:

1. Introduction; the authors should clarify why spinal surgery patients are particularly vulnerable to postop gout flare or at least differ from other type of surgery. Such explanation would help strengthen the rationale for conducting this study.

2. Methods: what is the variable 'medication for gout control'? Does this include urate-lowering therapy? The particular variable is also significant in the logistic regression model as a risk factor for flare. Therefore its definition should be clearly stated.

3. Methods: how did you manage patients who experience more than one episode of gout flare after surgery, as well as patients who underwent more than one operation within the study period?

4. Methods: Was sample size estimated and was it sufficient to support a robust regression model?

5. Methods: Could you provide reference for the formula for predicted odds? I am not sure how should we interpret the number (as well as its cut-off). It would be great to explain how predicted odds are different from the usual predicted probability?

6. Methods: Please explain how exposure variables are selected for inclusion to the final multivariate logistic regression model. I noticed that several other variables were collected but were omitted from the regression model.

7. Table 1 and 2: please include percentage to the tables for easier interpretation.

8. Table 3: please report regression coefficients (beta) for all variables in the regression models.

9. Table 5 is not relevant to the main discussion and should be removed.

10. Discussion (Line 195-198): blood transfusion and IV fluid has been found to be associated to in-hospital gout flare in general hospitalized patients (Ref: Jatuworapruk, K., et al. (2020). Ann Rheum Dis 79(3): 418-423.). This could be mentioned in the discussion.

Reviewer 2 Report

In this paper, the authors propose a prediction model for helping identify patients with higher risk of gout flare, and thus could be given optimal medical treatment. The paper is generally convincing and easy to follow. However, there are several concerns that need to be addressed.

       It would be more convincing if the authors can provide a validation data set to show how accurate the prediction is.

       There are still several grammar/spelling mistakes need to be corrected.

Round 2

Reviewer 1 Report

I have no further comments.